# Multilayer Nanofiber Composite Separator for Lithium-Ion Batteries with High Safety

**DOI:** 10.3390/polym11101671

**Published:** 2019-10-14

**Authors:** Wenxiu Yang, Yanbo Liu, Xuemin Hu, Jinbo Yao, Zhijun Chen, Ming Hao, Wenjun Tian, Zheng Huang, Fangying Li

**Affiliations:** 1College of Textile and Garment, Hebei University of Science and Technology, Shijiazhuang 050018, China; wenxiuyang-hbust@outlook.com (W.Y.); 13014375991@163.com (X.H.); 2School of Textile Science and Engineering, Wuhan Textile University, Wuhan 430200, China; 15532192903@139.com (Z.C.); 18533098646@139.com (M.H.); 15533605582@139.com (Z.H.); m15232132913@163.com (F.L.); 3School of Textiles, Tianjin Polytechnic University, Tianjin 300387, China; ywx880418@sina.com; 4School of Chemistry and Chemical Engineering, Wuhan Textile University, Wuhan 430200, China

**Keywords:** tipped needleless electrospinning, Von Koch curves spinneret, multilayer composite separator, high safety, lithium-ion battery

## Abstract

An original Von Koch curve-shaped tipped electrospinneret was used to prepare a polyimide (PI)-based nanofiber membrane. A multilayer Al_2_O_3_@polyimide/polyethylene/Al_2_O_3_@polyimide (APEAP) composite membrane was tactfully designed with an Al_2_O_3_@ polyimide (AP) membrane as outer shell, imparting high temperature to the thermal run-away separator performance and a core polyethylene (PE) layer imparts the separator with a thermal shut-down property at low temperature (123 °C). An AP electrospun nanofiber was obtained by doping Al_2_O_3_ nanoparticles in PI solution. The core polyethylene layer was prepared using polyethylene powder and polyterafluoroethylene (PTFE) miniemulsion through a coating process. The addition of PTFE not only bonds PE power, but also increases the adhesion force between the PE and AP membranes. As a result, the multilayer composite separator has high safety, outstanding electrochemical properties, and better cycling performance as a lithium-ion battery separator.

## 1. Introduction

With growing global energy demands, lithium-ion batteries (LIBs) are widely applied in energy storage systems and are expected to achieve high energy density, excellent electrochemical properties, and a long cycle life [1]. The LIB separator allows lithium ions to transport, playing a vital role in holding back cathode and anode contact thus avoiding the occurrence of short circuit [2]. In most commercialized LIBs, polyolefin-based membranes have been the most dominant over the decades, on account of their significant chemical stability and excellent mechanical properties [3]. Nevertheless, the defects of polyolefin-based microporous membranes, such as low porosity and insufficient electrolyte wettability, result in poor electrochemical properties of the batteries. Furthermore, inferior thermal-dimensional stability of LIB separators caused by polymers and lowering the softening point or melting point is regarded as a serious potential risk, primarily responsible for lithium-ion battery explosions [4]. Therefore, a LIB separator with high electrochemical performance and high safety is urgently required. 

Electrospinning is a simple and straightforward way to prepare a nanofiber membrane with high porosity [5]. Many polymers, such as PAN (Polyacrylonitrile) [6,7,8,9], PVDF (Polyvinylidene fluoride) [7,10,11], PVDF-HFP (Polyvinylidene fluoride-hexafluoropropylene) [12,13,14], and PI (Polyimide) [2,9], are used to prepare LIB membranes by electrospinning. Inorganic nanoparticles, such as TiO_2_ [15,16,17], SiO_2_ [18], Al_2_O_3_ [19], and ZrO_2_ [20], have been used to prepare composite lithium-ion battery separators, which can enhance the mechanical and electrochemical properties of the separator. Zhang et al. [21] prepared a PVDF/SiO_2_ composite membrane by dispersing different contents of silicon dioxide solution in 7% PVDF electrospinning solution. Then, a homogeneous electrospinning solution of SiO_2_ and PVDF was obtained. The results of PVDF/SiO_2_ membranes exhibit better thermal properties, high electrolytes, outstanding porosity, superior ionic conductivities, and excellent electrochemical performance. However, compared to polyolefin separators, the composite separators mentioned above could efficiently enhance the electrochemical properties, but could not achieve high safety levels without the addition of a polymer with a lower melting point, imparting a thermal shut-down function by blocking the pores. 

To improve the electrochemical properties and safety of LIBs, PI with a high melting point and PE with lower melting point were chosen as candidates for a LIBs separator. We previously designed a needleless electrospinning device based on Von Koch fractal curves, which demonstrated better field intensity and distribution, and high productivity compared to multineedle electrospinning after simulation and analysis on the electric field. Further simulation and analysis by finite-element analysis software showed that the circular type of fractal spinneret with the second-level fractal structure exhibited better field intensity and distribution compared to other nonlinear fractal electrospinning technology (Figure 1) [22]. The present study takes advantage of the original needleless electrospinning device and employs it for the fabrication of composite nanofiber membranes for a lithium-ion separator. Thanks to its outstanding thermal-dimensional stability and chemical stability, electrospun PI nanofibers were chosen to comprise the separator matrix [23]. Al_2_O_3_ nanoparticles mixed within polymer solution were applied to further enhance thermal-dimensional stability and electrochemical properties. PE particles layer with intrinsic lower melting features play an important role in blocking the lithium-ion conducting pathway at temperatures of about 123 °C. Thus, in this study, a three-layer composite sandwich structure of lithium-ion separator (Al_2_O_3_@PI/PE/Al_2_O_3_@PI, APEAP) was prepared through the thermal calendaring process, a PI membrane with Al_2_O_3_ nanoparticles (Al_2_O_3_@PI, AP) as the outer layer, and PE particles layer as the core layer (Figure 2). Compared with conventional polyolefin-based membranes, PI and AP, an APEAP separator exhibits excellent thermal-dimensional stability, preeminent electrolyte uptake, and superb electrochemical property. 

## 2. Experimental

### 2.1. Materials

Polyimide (PI, *M*_w_ = 150,000) was provided by Mitsui Chemical Co. Ltd., Japan (Tokyo, Japan). *N*-methyl-2-Pyrrolidone (NMP), ethanol, and n-butanol were purchased from Shanghai Siyu Chemical Technology Co., Ltd., Chain (Shanghai, China). Aluminum oxide (Al_2_O_3_) nanoparticles and polyethylene powders (PE) were obtained from Shanghai Naiou Nano-Science and Technology Co., Ltd., (Shanghai, China) Chain and Beijing Lishihang Co., Ltd., Chain, (Beijing, China) respectively. Polyterafluoroethylene (PTFE) miniemulsion was obtained from Dajin Industry Co., Ltd., Japan (Tokyo, Japan). Electrolyte solution 1M Lithium hexafluorophosphate (LiPF_6_) in ethylene carbonate (EC): diethyl carbonate (DEC) = 1:1(*v*:*v*) was provided by Tianjin Guangfu Chemical Reagent Co. Ltd., China, (Beijing, China).

### 2.2. Fabrication of APEAP Membranes

#### 2.2.1. Fabrication of PE Membrane

The PE slurry was obtained by PE powders, PTFE miniemulsion, and ethanol mixed in the ratio of 9:1:18 (w:w:w), after ultrasonic treatment for 0.5 h at room temperature. A PE microporous membrane was prepared using Elcometer 4340 Automatic Film Applicator. The scraper thickness was 20 μm, and the speed of the scraper was 3 cm s^−1^. After the coating process, the prepared PE microporous membrane was dried in a vacuum at 70 °C for 12 h to remove the solvent.

#### 2.2.2. Fabrication of PI and AP Electrospun Membrane

PI and AP membranes were prepared by self-assembly Von Koch-based tipped needleless electrospinning apparatus. The circle spinnerets with Level 2 fractal structure were used as the spinning electrode of the electrospinning apparatus and made of stainless steel. The inner diameter of the spinnerets was 69 mm, the outer diameter was 79 nm and the thickness was 2 mm. More details about the device can be found in our previous work [22]. PI electrospinning solution (30% by wt.) was made by dissolving 3 g of PI particles in 7 g of NMP with magnetic stirring for 5 h at a temperature of 80 °C. AP solution was obtained by mingling Al_2_O_3_ nanoparticles into the PI solution mentioned above at ratios of 1% and 2% by wt. of PI, respectively, with vigorous stirring for 5 h at 80 °C, followed by 1 h ultrasonic treatment (30% by wt. PI solution without adding Al_2_O_3_ inorganic nanoparticles was noted as 0% AP). To prepare the membrane, the electrospun voltage was 20 kV, the rotation speed of the spinning electrode was 90 rpm, and the needle-to-collector distance was 20 cm. After being deposited, the freestanding fiber membrane was peeled from the release paper. 

#### 2.2.3. Fabrication of APEAP Membrane

The multilayer APEAP composite membranes were composed of AP membrane sheath layers and PE microporous membrane as the core layer through a thermal calendaring process by a pneumatic double-station semi-automatic pressing machine (Model: JDS-SHT, provided by Shenzhen Hongtai Silk-Printing Equipment Factory, Shenzhen, China) under 0.8 MPa and at 131 °C for 40 s. The LIBs were assembled in a glove box (Shanghai Tianhe Purification Cycle Glove Vox (Shanghai) Co., Ltd., Shanghai, China). The whole preparation process flow diagram of the multilayer composite separator is illustrated in Figure 2. 

### 2.3. Characterization 

The surface morphologies of all the membrane samples previously including PE, before and after heating, PI, AP, and APEAP, with cross-section and energy dissipation spectroscopy (EDS), were evaluated using field emission scanning electron microscopy (FESEM, S-4800, Hitachi, Tokyo, Japan) and Fibermetric software (PHENOM PROX Desktop FESEM), after separators were sputter-coated with gold. The fiber diameter was measured by Image Pro Plus software. A transmission electron microscope (TEM, JEM 2010, Tokyo, Japan) was used to demonstrate the existence and distribution of Al_2_O_3_ nanoparticles in the PI composite nanofibers. Electrospun conductivity and viscosity was measured using the a meter (FE30, METTLER-TOLEDO Instruments Shanghai Co., Ltd., Shanghai, China) and a rotor flow meter (NDJ-8S/5S, Li Chen technology co. LTD Co., Ltd., Beijing, China), respectively.

The porosity (P) of the membranes were investigated by dipping them into n-butanol for 2 h, and then the porosity could be obtained by following Equation (1): (1)$$P\;=\;\frac{\frac{m_{w}}{\rho_{w}}}{\frac{m_{w}}{\rho_{w}}+\frac{m_{d}}{\rho_{d}}}\times 100\%$$
where *m_w_* and *m_d_* represent the wet weight and dry weight, respectively. *ρ_w_* and *ρ_d_* refer to the density of the solvent (n-butanol) and PI, respectively.

Liquid electrolyte uptake was performed by weighing the mass of samples before and after dipping in the electrolyte solution. Then, the liquid electrolyte uptake could be obtained by following Equation (2): (2)$$Uptake\%=\frac{\left(W-W_{0}\right)}{W_{0}\times 100\%}$$
where *W* and *W*_0_ represent the dry weight and wet weight, respectively.

The contact angle measurement was carried out by dropping the electrolyte solution on the surface of the sample using a contact angle-measuring instrument (SDC-100, Beijing Dongfang Defi Instrument Co. Ltd., Beijing, China). The contact angle was measured immediately after that the electrolyte dropped onto the membrane.

The thickness of the separators was measured by a CHY-C2 thickness tester. Mechanical strength and peeling strength was tested by an Instron 3369 Universal Strength Tester at constant rates of 20 mm s^−1^, 50 mm s^−1^, with specimen dimensions of about 2 cm × 10 cm and 2.5 cm × 7.5 cm, respectively. The peeling strength of PE membrane coated onto the electrospun membrane was evaluated using the ASTM D 2261-13 standard method.

The thermal property of the separators was obtained by DSC measurement using a TA DSC2920 tester. To eliminate the influence of impurities and water, a secondary heating curve was used. Under a nitrogen atmosphere, the temperature was raised to 150 °C at a rate of 10 °C/min. Then, it was cooled to room temperature at a rate of 20 °C/min. Finally, the temperature was raised to 300 °C at a rate of 10 °C/min.

Thermogravimetric analysis (DTG-60H, Shimadzu, Kyoto, Japan) was employed to evaluate the thermogravimetrics of various composite membranes. The measurement was performed at 30–800 °C with a heating rate of 10 °C/min. Nitrogen was flushed at 70 mL/min. Derivative thermogravimetry (DTG) was carried out to evaluate the temperature of the maximum mass-loss rates.

Ionic conductivity and interfacial stability were investigated by EIS (Electrochemical impedance spectrum) using a Zahner Zennium electrochemical analyzer (CHI660D electrochemical workstation, Shanghai Chenhua Instrument Inc., Shanghai, China) over a frequency ranging from 0.1 to 1 MHz, with 5 mV of AC amplitude. The ionic conductivity and interfacial stability were obtained using the apparatus mentioned above. The ionic conductivity was obtained by following Equation (3): (3)$$\sigma =\frac{b}{R_{b}\times A}$$

In this equation, σ refers to the ionic conductivity. *R_b_* represents the bulk resistance measured from AC impedance test. *b* and *A* refer to the thickness and area of the samples.

The electrochemical stability window was also carried out using the half-cell, which is composed of a stainless-steel electrode acting as the working electrode, and a lithium metal acting as a counter electrode. An electrochemical analyzer CHI 660D (Shanghai Chenhua Instrument Inc., China) was used to record the electrochemical stability window under a scanning rate of 10 mV s^−1^ and over a potential ranging from 0 to 6 V using the linear sweep voltammetry method at room temperature.

Electronic conductivity was evaluated by SDS50 Dielectric Spectrometer provided by Novocontrol GmbH. The charge–discharge tests of the cells containing liquid electrolyte-soaked membranes were measured by the LANHE battery test system (Wuhan Blue Electric Co. Ltd., Wuhan, China), under potentials ranging from 2.8 to 4.2 V, at 0.2 C rate.

## 3. Results and Discussion

### 3.1. Nanofiber Morphology and Electrospun Solution Property

The surface morphology and its solution property of the electrospun membranes containing various contents of Al_2_O_3_ nanoparticles (0%, 1%, 2%, 3% by wt. of PI) and the PI fiber diameter distributions are shown in Table 1 and Figure 3a–d, respectively. The conductivity of the electrospun solution without nanoparticles is 1.37 μm/cm. The conductivity of the electrospun solution increased with the addition of nanoparticles. The increasing of conductivity leads to an increase in the charge density of the polymer chains, which strengthens the repulsive force between the polymer chains, during a process of nanofiber formation. The viscosity of the electrospun solution without nanoparticles is 1.29 Pa·s. The viscosity of the electrospun solution decreased with the addition of nanoparticles. The decrease of viscosity results from the incorporation of Al_2_O_3_ nanoparticles, which decrease the intermolecular forces of the polymers during the electrospinning process [24,25,26]. It can be seen that from Table 1 and Figure 3 that the diameter of the PI fiber without nanoparticles ranges between 800 nm and 1400 nm, with an average of 1125 nm. With the increase in the amount of Al_2_O_3_ nanoparticles, the average diameters of PI nanofibers with 1%, 2%, and 3% nanoparticles decrease to 914 nm, 524 nm, and 401 nm, respectively, which is attributed to the increased conductivity and the decreased viscosity of the electrospun solution. A solution with high conductivity has much greater charge-carrying capacity than a solution with low conductivity, which will make the polymer jet formed by the high-conductivity solution receive greater drafting force under the action of electric field force, and the fiber is also thinner. With lower viscosity, the movement ability of polymer chain segments is limited, and it is easier to be drawn under electric field force, resulting in thinner fiber diameter.

However, as observed from Figure 3 and Table 1, beaded fibers are obviously produced, when nanoparticle concentration increases to 3%, which may be caused by the agglomeration of nanoparticles [27]. Thus, the result of SEM images shows that nanofibers with 2% nanoparticle concentration have smaller diameters and a better fiber morphology. The sample AP membrane containing 2% Al_2_O_3_ displayed a morphology with a smooth surface and the finest fiber size (524 nm diameter), as well as reasonable diameter distribution (20.45%), as well as optimal fiber morphology compared to other AP samples. This sample can be used for later discussion and analysis.

After the thermal calendaring process at a temperature of 123 °C for 60 s, under pressure of 0.8 MPa, the thickness of the PE particle layer is 14 μm, the thickness of the electrospun AP membrane is 11 μm, and the total thickness of the composite separator is 36 μm. Figure 3f shows the presence and uniform distribution of Al_2_O_3_ nanoparticles in PI nanofibers.

### 3.2. Wettability

Porosity and electrolyte uptake are two important indicators of the wettability of the separator and the results of porosity and electrolyte uptake are presented in Table 2 [28]. The porosities of PI, AP, and APEAP separators are superior to the PP/PE/PP separator, which is because of the 3D network structure of the electrospun membrane. The porosity of the APEAP separator (72%), higher than the PI membrane (64%), is observed with the addition of the Al_2_O_3_ nanoparticle as shown in Table 2. The improvement to porosity owing to the presence of Al_2_O_3_ nanoparticle is caused by a thinner fiber diameter. The porosity of the separator is powerfully dependent on the size of the nanofiber; a thinner fiber diameter separator can form more void space [29]. The porosity of the APEAP separator as the combination of AP and PE separators is between the two kinds separator.

The electrolyte uptakes of PP/PE/PP, PI, AP, and APEAP separators are shown in Table 2. Benefiting from the micropore structure of electrospun nanofibers membrane, the electrolyte uptakes of PI, AP, and APEAP membranes are much higher than the PP/PE/PP separator. Except for the reasons mentioned above, the similar polarity between the PI nanofiber-based separator and the liquid electrolyte also guarantees a superior electrolyte uptake for the PI nanofiber-based separator [30]. In addition, the capillary effect of the nanofibers leads to the nanofiber separator having outstanding electrolyte uptake [31]. Relying on the preponderance mentioned above, the electrolyte uptake of PI, APEAP, and APEAP separators is higher than 400%. It can be observed that the porosity of the AP separator increases with the addition of nanoparticles, on account of the higher porosity of the AP separator compared to the PI separator, which can absorb more liquid electrolyte. Although the electrolyte uptake decreased because of the addition of PE component, the electrolyte uptake of the APEAP separator is still as high as 592%. Compared with other work, the electrolyte uptake of APEAP is also exciting (Appendix A).

The contact angle of the electrolyte plays a very important role in battery properties, thanks to the close relationship with the electrochemical property of LIBs. The better wettability of the separators could promote electrolyte permeation during battery assembly. Hence, having satisfied wettability for electrolytes, separators could be a favorable media for the transport of lithium ions during the charging and discharging process, which will contribute to the electrochemical performance of LIBs. The contact angles of various membranes are shown in Table 2. It can clearly be seen that the contact angles of PI-based membranes were lower than PP/PE/PP membranes. The lower contact angle of PI-based membranes are attributed to its porous structure formed by nanofibers, and the capillary effect of nanofibers on the surface of the membrane. In addition, the non-polar PP/PE/PP membrane has inherent poor wettability with the electrolytes. Compared to the PI separator, the AP separator has better wettability attributed to its higher porosity and superior electrolyte uptake. Moreover, the Al_2_O_3_ nanoparticles lying on the surface of the AP nanofiber also contributed to aggrandizing electrolyte wettability because of the affinity of the Al_2_O_3_ to electrolyte components. There is no difference between the contact angle of the AP and APEAP separator because the AP and APEAP separators have the same material on the surface.

### 3.3. Mechanical Performance

The typical stress–stain curves are shown in Figure 4. PP/PE/PP has excellent mechanical properties, and the tensile strength of the PP/PE/PP membrane is as high as 150.34 MPa, as Figure 4 demonstrates. The tensile strengths of the PI, AP, and APEAP separators are 5.9 MPa, 7.9 MPa, and 9.5 MPa, respectively. The mechanical property of AP is better than PI, for the addition of the nanoparticles increasing the number of the intersections among fibers, and then improving the friction force. The APEAP separator has the highest tensile strength compared to PI and the APEAP separator, except for the reason mentioned above; at the APEAP separator, the adhesion force between the PE and AP membrane is another important factor.

In addition, the peeling strength of the PE layer to the PI electrospun nanofiber membrane between the surfaces of AP nanofiber layer and PE membrane is evaluated by conducting peel tests on the Instron Tensile Tester, peeling the PE layer away from the AP membrane. The resultant peel strength in Figure 4a indicates that the PE surface produced good adhesion with the electrospun membrane, with peel strength being 5.6 MPa. Consequently, the above result indicates that the APEAP composite membrane is equipped with robust mechanical performance, which could be used as a promising material for potential applications in LIBs. The reason for excellent adhesion strength can be explained by the following: (1) through a thermal calendaring process at the melting point, under 0.8 MPa, the fusing PE particles on the surface of PE membrane enhanced the adhesion strength between the PE membrane and the AP membrane; (2) PTFE may be used as a binder to improve the bonding strength between PE particles.

### 3.4. Thermal-Dimensional Stability

The surface morphology of the PE membrane before and after heat treatment at 123 °C for 30 min is shown in Figure 5a,b. From the images, it can be observed that before heat treatment there are a lot of micropores on the surface of PE membrane. After heat treatment, micropores are blocked by the fusion of the PE particles at a temperature of 123 °C, realizing the function of thermal shut-down. It can be observed that from Figure 5c, the DSC curves of the PE and APEAP membranes, an endothermic peak occurred at 123 °C, representing the melting point of the PE component.

The thermal-dimensional stability of PP/PE/PP, PI, AP, and APEAP membranes, a significant factor to ensure battery safety performance, was measured by hot oven test at 180 °C for 0.5 h, and the results are presented in Figure 6. The PP/PE/PP separator could not maintain dimensionality at 180 °C and dimensional change is obvious. Compared with the commercial PP/PE/PP separator, the APEAP separator exhibited outstanding thermal-dimensional stability, as Figure 6 shows. The outstanding thermal-dimensional stability attributed to the heat-resistant AP separator served as a skeleton to guarantee intact structure. In addition, the inorganic nanoparticle also plays the function of heat-resistant materials to maintain thermal-dimensional stability. Thermogravimetric analysis and derivative thermogravimetry (DTG) also show that the APEAP composite membrane has better thermal stability than the commercial PP/PE/PP membrane.

### 3.5. Electrochemical Performance

Figure 7a shows the Nyquist curves of the liquid electrolyte-soaked membranes. The high-frequency intercept of the Nyquist curve on the real axis represented the bulk resistance of the separator. As Figure 7a and Appendix A shows, PI nanofiber-based separators have excellent ionic conductivity than a polyolefin-based separator. Between them, the APEAP separator has the highest ionic conductivity. Compared with the PI and AP separator, the AP separator has higher ionic conductivity attributed to the adding of the nanoparticles, leading to AP separators possessing higher electrolyte uptake and fully interconnected micropores. The ionic conductivity of the polyolefin-based separator is worst, owing to its lowest porosity and inferior wettability.

Figure 7b and Appendix A shows the interfacial resistance of separators (EIS) of Li/liquid electrolyte-soaked membrane/Li cells. In these EIS curves, the diameter of the semicircle at the medium to low region indicates the electrolyte interfacial resistance. The order of interfacial resistance values is: PP/PE/PP (99 Ω) > PI (78 Ω) > APEAP (69 Ω) > AP (62 Ω). The PI, APEAP, and AP separators show lower interfacial resistance than the commercial PP/PE/PP separator, which is attributed to poor porosity, weak electrolyte uptake, and inferior membrane–electrode affinity of the commercial PP/PE/PP separator. The cells assembled with the AP separator exhibited lower interfacial impedance than the PI separator, indicating that the lithium ion on the electrode of the AP battery can more easily pass through the electrolyte film than the lithium ions on the electrode of the PI battery. It also can be seen that there is no significant difference between the interfacial impedance of AP and APEAP separator, manifested by there being no influence of the presence of PE membranes between the AP membranes on the interfacial resistance.

The electrochemical stability window is an important indicator to measure the stability of the electrolyte. The electrochemical stability window of liquid electrolyte-soaked separator is obtained by linear sweep voltammetry (LSV) in the system of Li/separator/stain steel cells ranging from 0 to 8 V at a scan rate of 10 mV s^−1^. Within the scope of the electrochemical stability window, there is no redox reaction, the battery can work normally over the electrochemical stability window, and the electrolyte incurred oxidative decomposition. The electrochemical stability window of liquid electrolyte-soaked membranes are shown in Figure 7c and Appendix A. The electrochemical stability windows of the PI, AP, and APEAP separators are much better than PP/PE/PP, which can be attributed to the higher porosity, superior electrolyte uptake, and better wettability with the liquid electrolyte. The AP separator exhibited higher decomposition voltage, beneficial when more liquid electrolyte enters the pores of the fibrous separator, which demonstrates that the AP separator possesses better electrochemical stability. As a result of the existence of the PE membrane in the APEAP separator, the electrochemical stability window of APEAP is only slightly smaller than the AP separator, and still at a higher level.

### 3.6. Battery Performance

For the sake of estimating the feasibility of adopting PI, AP, and APEAP membranes in rechargeable lithium-ion batteries, the electric conductivities of separators were measured by an SDS50 Dielectric Spectrometer. The electric conductivities of PI, AP, and APEAP membranes and PP/PE/PP membranes for comparison are 8.27 × 10^−15^ S cm^−1^, 8.58 × 10^−14^ S cm^−1^, 4.67 × 10^−14^ S cm^−1^ and 9.96 × 10^−14^ S cm^−1^ under the frequency of 0.01 Hz. Although the electric conductivities of AP, and APEAP separators are higher than the PI separator, they are still lower than the PP/PE/PP membrane, and all kinds of separator have the property of electric insulation. Therefore, the PI, AP, and APEAP membranes can be used for LIBs without potential safety problems.

The coin-type cells that use various membranes are assembled with LiCoO_2_ and metal lithium. The initial charge–discharge of cell capacity assembled with PP/PE/PP, PI, AP, and APEAP membranes at a current density consistent with 0.2 C and cycled under a voltage between 2.8–4.2 V are shown in Figure 8. Many researchers have discovered that the type and structure of the separators also play a significant role in deciding the cell performance because the type and the structure of the separator influences the lithium-ion transportation between the electrodes [32]. The initial discharge capacity of cells with commercial a PP/PE/PP separator is 125.6 mAh g^−1^. Relying on hydrophilicity of liquid electrolyte uptake and a 3D network structure with high porosity, the initial discharge capacities of the Li/LiCoO_2_ cells with PI, AP, and APEAP separators are higher than the PP/PE/PP separator. The order of the initial discharge capacities of the Li/LiCoO_2_ cells with PI, AP, and APEAP separator is coincident with the orders of the liquid electrolyte and the porosity. The AP separator possesses higher liquid electrolyte uptake and greater porosity, and as a result, has a better ionic conductivity, which provides more channels to transport more lithium ions between electrodes and accelerates the intercalation and de-intercalation of lithium ions in and out of the electrode materials, giving rise to increased discharge capacity [33]. It is worth mentioning that the initial discharge capacity of the cell with the thermal shut-down-function APEAP separator ranks second to the cells with AP separator, and has a better level.

The cycling performance of cells with PP/PE/PP, PI, AP, and APEAP separators are measurement by evaluating the discharge capacities of all cells with the corresponding separator; the composition of the cells is the same as the first charge–discharge tests. The cycling performance of all Li/LiCoO_2_ cells at 0.2 C rate under constant current conditions is shown in Figure 8. For the cell assembled with a PP/PE/PP separator, there is serious capacity loss and the discharge capacity after 100 cycles reduces from 125.6 mAh/g to 80.7 mAh g^−1^ with poor capacity retention of 64.3%. The capacity loss during the cycling process could be brought about by unwished side reactions caused by the decomposition of the liquid electrolyte [32]. Accompanying the decomposition of the liquid electrolyte, active material dissolution, a passive film formation over the electrode material, and as a result, decreased the active material of the electrodes caused the capacity loss [2,34]. The cells with PI, AP, and APEAP separators show excellent cycling performance after 100 cycles (Appendix A). The outstanding cycling performance depends on strong affinity for the liquid electrolyte and a highly porous structure, which could impart easier lithium-ion transport and excellent liquid electrolyte retention during the cycling process [35]. The cycling performance of cells with AP and APEAP separators are attributed to inorganic nanoparticles improving the interface stability between the electrodes and the electrolyte, reducing the formation of the passive film, and cutting down the capacity retention. Hence, the APEAP with thermal function is a better choice as a lithium-ion separator [36].

## 4. Conclusions

A thermal shut-down function sandwich-structured composite APEAP membrane used as LIB separator was fabricated by electrospinning, coating, and thermal calendaring technique. The multilayer separator was fabricated with two layers: an electrospun layer AP as outer, and one layer of coating PE as core. Compared to the PP/PE/PP separator, the existence of an electrospun membrane enhanced the liquid electrolyte, porosity, and wettability of the membrane, owing to the high specific surface area. The addition of nanoparticles in the electrospinning solution decreased the nanofiber average diameter from 913 nm to 561 nm. The good mechanical properties and outstanding thermal-dimensional stability greatly improved the safety of the LIBs. Furthermore, excellent ionic conductivity, interfacial stability, and electrochemical stability, as well as superior battery cycling performance also proved that electrospun membranes with nanoparticles are applicable for a lithium-ion battery. Furthermore, it is important to add a PE layer giving the composite APEAP separator a thermal shut-down function with little loss of electrochemical and cycling performance. Therefore, sandwich-structure composite APEAP membrane is the ideal choice for a high-performance LIB separator.

## Figures and Tables

**Figure 1 polymers-11-01671-f001:**
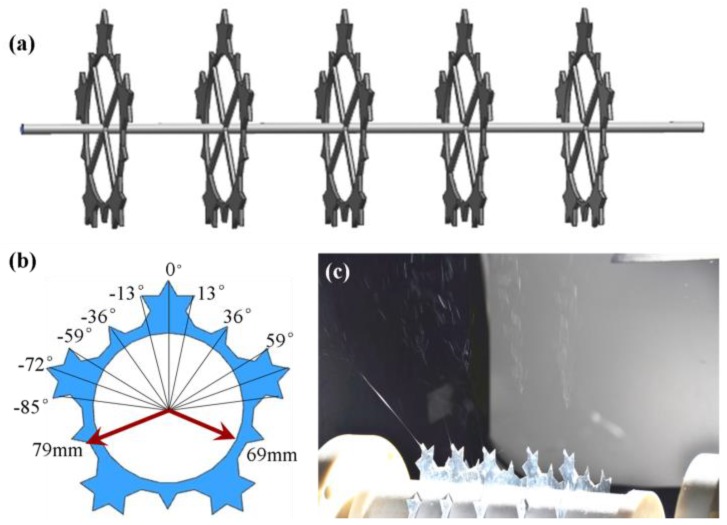
The Von Koch curve-shaped tipped electrospinneret; (**a**) The Von Koch curve-shaped tipped electrospinneret model; (**b**) the specification of the Von Koch electrospinneret; (**c**) the needleless electrospinning process.

**Figure 2 polymers-11-01671-f002:**
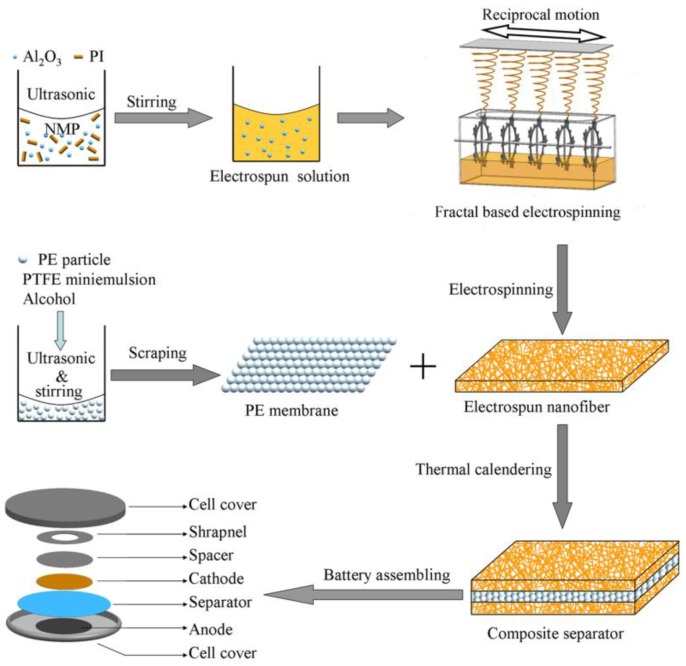
The flowchart of the fabrication of multilayer composite membrane.

**Figure 3 polymers-11-01671-f003:**
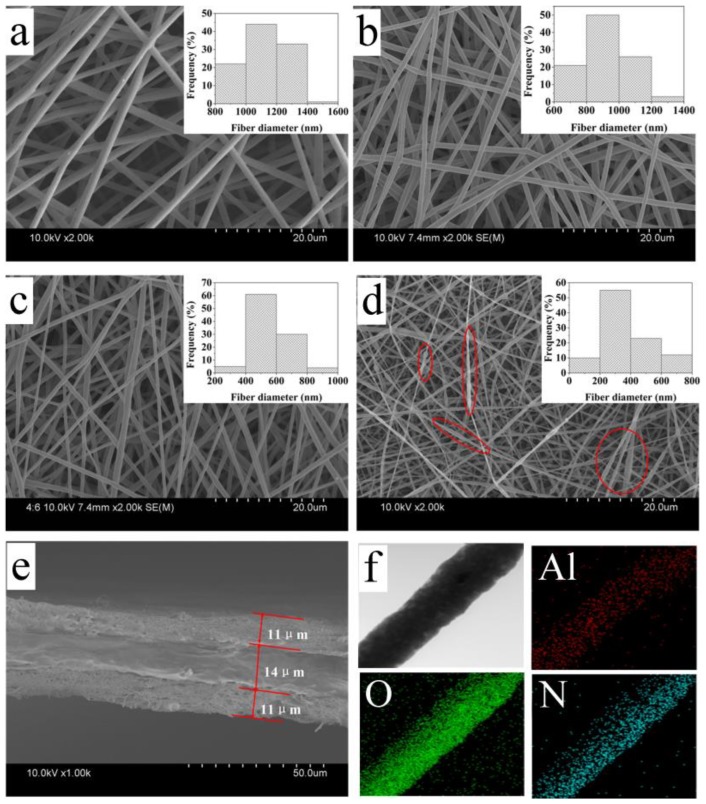
The SEM images of PI with different nanoparticle concentration of (**a**) 0%, (**b**) 1%, (**c**) 2%, (**d**) 3%, (**e**) cross-section image of APEAP separator, (**f**) HAADF-STEM (high-angle annular dark field-scanning transmission electron microscopy), and the corresponding EDS (energy dispersive spectrometer) mapping of the Al, O, and N elements in composite nanofibers.

**Figure 4 polymers-11-01671-f004:**
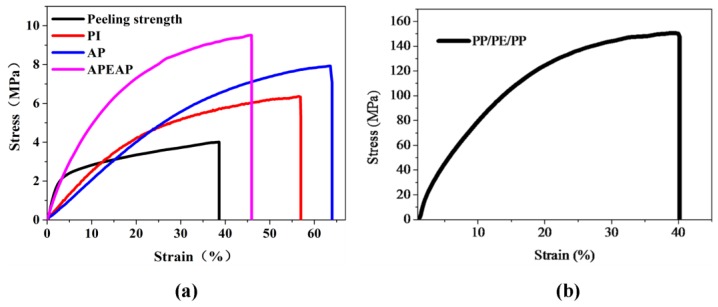
The stress-strain curves of various separators and the adhesion strength between PE and AP membrane: (**a**) The stress-strain curves of PI based separators; (**b**) The stress-strain curve of PP/PE/PP separator.

**Figure 5 polymers-11-01671-f005:**
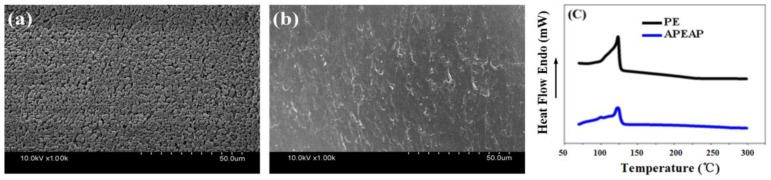
The SEM image of PE membrane before and after heat treatment and DSC curves of PP/PE/PP, PE, and APEAP membrane: (**a**) before, (**b**) after, (**c**) DSC curves of PE and APEAP membrane.

**Figure 6 polymers-11-01671-f006:**
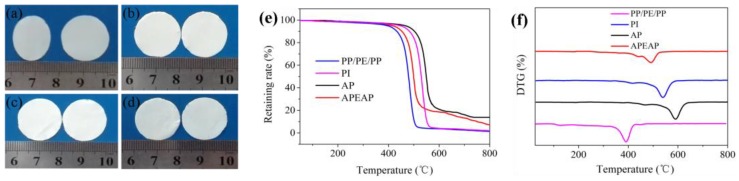
Photograph of separators before and after thermal treatment at 180 °C for 0.5 h: (**a**) PP/PE/PP; (**b**) PI; (**c**) AP; (**d**) APEAP; Thermogravimetric Analysis (TGA) (**e**) and Derivative Thermogravimetry (DTG) (**f**) of various membranes.

**Figure 7 polymers-11-01671-f007:**
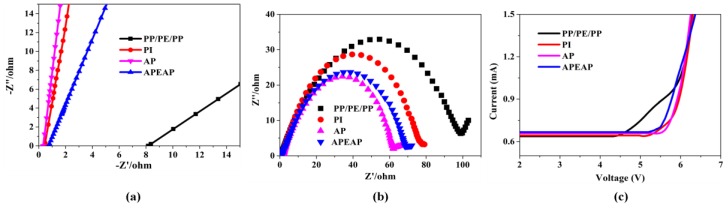
Electrochemical performance of different separators: (**a**) Impedance spectra of the PP/PE/PP, PI, AP, and APEAP separator; (**b**) The AC impedance spectra of the half-cell with different separators; (**c**) Linear sweep voltammograms of different separators.

**Figure 8 polymers-11-01671-f008:**
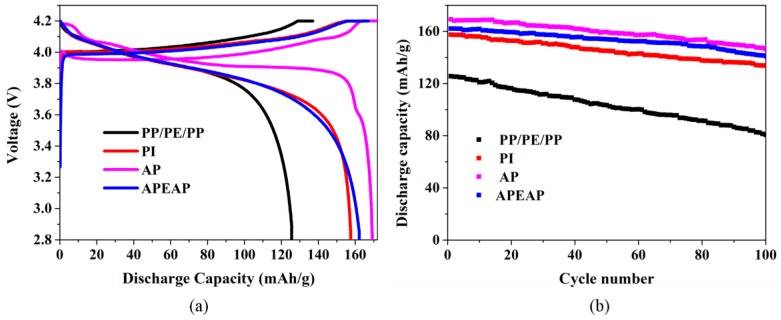
Cycling performance of different Li/LiCoO2 cells containing various separators at 0.2 C: (**a**) First charge-discharge curves of different Li/LiCoO2 cells containing; (**b**) The 100 times charge-discharge curves.

**Table 1 polymers-11-01671-t001:** Morphology change in PI nanofibers after addition of Al_2_O_3_ nanoparticles.

Al_2_O_3_ % Against PI (by wt.)	Conductivity (μm/cm)	Viscosity (Pa·s)	Nanofiber Diameter	
			Average (nm)	CV %
0	1.37	1.29	1125	12.17
1	15.68	0.92	914	14.17
2	19.49	0.59	524	20.45
3	35.82	0.25	401	51.56

**Table 2 polymers-11-01671-t002:** Porosity, electrolyte uptake, and contact angle of the samples.

Sample ID	PP/PE/PP	PI	AP	APEAP
Porosity (%)	39	64	78	72
Electrolyte uptake (%)	92	467	611	592
Contact angle (°)	53	24	19	19

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
