# Peer review of "Multilayer Nanofiber Composite Separator for Lithium-Ion Batteries with High Safety"

_polymers, 2019, doi:10.3390/polym11101671_

Round 1

Reviewer 1 Report

The topic is very interesting, the basic idea is concerning the methodology well implemented and the methods used are appropriate. The overall structure is understandable and the conclusions essentially comprehensible.
However, the manuscript is difficult to understand linguistically. There are many spelling, grammar and formatting errors throughout the manuscript. Many details have been insufficiently worked out. In some places it would make sense to compare the own results with the state of research by naming further sources. The evaluation of the results could have been deepened.

Introduction

Benefits of Van Koch curved shape and differences to other spinnerets should be explained briefly. Is it of importance to the substance of this paper? The reason for the chosen sequence of layers should be addressed.

Experimental

line 68: What specific “alcohol”? Ethanol? What is “polyethylene power”? Powder? “LiPF61mol L‐1, EC∶DEC=1∶1” abbreviation should be explained and written properly. LiPF6 1 mol/L? Has a moving collector been used or a static? What current?

There are several important details about setup and parameters of the devices used missing.

What thickness did the sputter coating have?

After how much time has the contact angle been measured?

Results and discussion

Not every number which is already in the table has to be restated.

Analysis of results in table 1 could be more extensive. “attributed to the increased conductivity and the decreased viscosity of the electrospun solution.” please elaborate on that.

“which is attributed to the agglomeration of nanoparticles” Here I would use the subjunctive. These nanofibers are not really "beaded" but rather rough, which could also be due to the reduced fiber diameter. As a result, significantly more fibers can be seen on the same field of view, and the areas marked in Figure 3 are selected very selectively or are not necessarily representative.

page 7: The high uptake is not surprising. Some comparable results from literature could have been interesting.

“There is no difference between the contact angle of the AP and APEAP separator, because the AP and APEAP separators have the same surface structure.” they not only have the same surface structure. It’s the same Material, isn’t it?

“PTFE is a binder of the PE particles, there are some PTFE molecules on the surface of PE membrane, and the presence of PTFE molecules increase the adhesion strength.” Please elaborate on this. It is hard to imagine that PTFE forms intermolecular bonding to increase adhesion.

Figure 5: No process parameters of DSC have been mentioned. What is the heating rate? Nitrogen atmosphere? Are the shown curves from the first heating cycle and why? The second cycle often obtains valuable information, too. The direction of endothermal and exothermal heat flow should be seen on the y axis.

“separators exhibited outstanding thermal stability without thermal shrinkage regardless of whether nanoparticles added or not,” the term “outstanding” could only be used in comparison with other options. “outstanding” compared to what?

page 9: “thermal stability” is not properly defined here. Is it dimensional stability? Or decomposition Temperature? Thermogravimetric Analysis could be helpful here?

Electrochemical properties could better be shown in a table.

Throughout the manuscript the formatting has been completely arbitrary: placement of space characters, paragraphs, punctuation marks, fonts and resolution in figures…

page 10 “show lower interfacial resistance than commercial PP/PE/PP separator”. Source?

“As we all know, the cell capacity …” This expression does not belong here.

Reviewer 2 Report

Paper „Multilayer Nanofiber Composite Separator for Lithium Ion Batteries with High Safety" contains information about preparation of the membrane used as separator for Li Ion Batteries. In my opinion the paper could be published in POLYMERS but should be improved.

1/ Introduction should be improved because the aim of the studies is not clearly specified.

2/ Figures of EDS spectra should be improved - quality of them are poor.

3/ The disscussion of the contact angle of the electrolyte should be done better. Information that contact angle plays very important role in battery properties should be developped.

4/ Description of the Al2O3 nanoparticles preparation should be shortly described.

Round 2

Reviewer 1 Report

Teflon is a trade name, the polymer is called polytetrafluoroethylene.

It is still unclear how PTFE can act as adhesion promoter (L278 P9). When speculating about causes, it should be clear that it is speculation rather than a fact.

The manuscript still contains plenty of spelling and grammatical errors. I would strongly recommend that you have a native speaker proofread. There is a space between the number and the unit - please revise it consistently throughout the manuscript. Similar minor mistakes are made throughout the document.

Reviewer 2 Report

The paper "Multilayer Nanofiber Composite Separator for Lithium Ion Batteries with High Safety" could be published in current version.

Author Response

Dear Reviewer

Thank you for reading our manuscript and reviewing it, which will help us improve it to a better scientific level.If there are still problems with our work, please don’t hesitate to tell us.